# Cell-Projection Pumping of Fibroblast Contents into Osteosarcoma SAOS-2 Cells Correlates with Increased SAOS-2 Proliferation and Migration, as well as Altered Morphology

**DOI:** 10.3390/biom11121875

**Published:** 2021-12-14

**Authors:** Swarna Mahadevan, James A Cornwell, Belal Chami, Elizabeth Kelly, Hans Zoellner

**Affiliations:** 1The Cellular and Molecular Pathology Research Unit, Oral Pathology and Oral Medicine, School of Dentistry, Faculty of Medicine and Health, The University of Sydney, Westmead Hospital, Westmead, NSW 2145, Australia; smah9946@uni.sydney.edu.au (S.M.); cornwellja@mail.nih.gov (J.A.C.); belal.chami@sydney.edu.au (B.C.); elizabeth.kelly@sydney.edu.au (E.K.); 2Laboratory of Cancer Biology and Genetics, Center for Cancer Research, National Cancer Institute, Bethesda, MD 20892, USA; 3Molecular Biomedicine, Charles Perkins Centre, The University of Sydney, Sydney, NSW 2006, Australia; 4Biomedical Engineering, Faculty of Engineering, The University of Sydney, Sydney, NSW 2006, Australia; 5Graduate School of Biomedical Engineering, University of NSW, Sydney, NSW 2052, Australia; 6Strongarch Pty Ltd., Sydney, NSW 2000, Australia

**Keywords:** fibroblast, cell-projection pumping, cancer cell proliferation, cancer cell migration, cancer cell morphology, cancer cell diversity, single-cell tracking

## Abstract

We earlier reported that cell-projection pumping transfers fibroblast contents to cancer cells and this alters the cancer cell phenotype. Here, we report on single-cell tracking of time lapse recordings from co-cultured fluorescent fibroblasts and SAOS-2 osteosarcoma cells, tracking 5201 cells across 7 experiments. The fluorescent lipophilic marker DiD was used to label fibroblast organelles and to trace the transfer of fibroblast cytoplasm into SAOS-2 cells. We related SAOS-2 phenotypic change to levels of fluorescence transfer from fibroblasts to SAOS-2 cells, as well as what we term ‘compensated fluorescence’, that numerically projects mother cell fluorescence post-mitosis into daughter cells. The comparison of absolute with compensated fluorescence allowed us to deduct if the phenotypic effects in mother SAOS-2 cells were inherited by their daughters. SAOS-2 receipt of fibroblast fluorescence correlated by Kendall’s tau with cell-profile area and without evidence of persistence in daughter cells (median tau = 0.51, *p* < 0.016); negatively and weakly with cell circularity and with evidence of persistence (median tau = −0.19, *p* < 0.05); and very weakly with cell migration velocity and without evidence of persistence (median tau = 0.01, *p* < 0.016). In addition, mitotic SAOS-2 cells had higher rates of prior fluorescence uptake (median = 64.9 units/day) than non-dividing cells (median = 35.6 units/day, *p* < 0.016) and there was no evidence of persistence post-mitosis. We conclude that there was an appreciable impact of cell-projection pumping on cancer cell phenotype relevant to cancer histopathological diagnosis, clinical spread and growth, with most effects being ‘reset’ by cancer cell mitosis.

## 1. Introduction

We earlier described the exchange of membrane, organelles and cytoplasmic protein between cultured human fibroblasts and cancer cells [1,2,3] and others have made similar observations [4,5,6,7,8,9,10,11,12,13,14,15,16,17,18,19,20,21,22,23,24,25,26,27]. The uptake of cellular contents by such transfers may be via exosomes, tunnelling nanotubes, or a mechanism we recently reported and termed ‘cell-projection pumping’ and these can significantly change the phenotype of acceptor cells [2,3,4,5,6,7,8,9,10,11,12,13,14,15,16,17,18,19,20]. The transfer of mitochondria appears especially significant and can confer chemotherapy resistance to cancer cells [4,6,7,20,21,22,23,24,25,26,27]. However, we are made wary of ascribing most phenotypic effects to mitochondrial exchange by our observation that, in addition to mitochondria, there is also bulk transfer of cytoplasmic proteins, plasma membrane-bound alkaline phosphatase and organelles smaller than mitochondria [1,2,3].

Exosomes comprise membrane-bound vesicular structures shed by cells, that are readily taken up by neighbouring cells [11,12,13,17,18]. Tunnelling nanotubes are delicate tube-like structures that establish cytoplasmic continuity between often distant cells and, in two-dimensional cell cultures, appear as pipe-like structures suspended above the culture surface [8,9,10,14,15,28,29,30]. Cell-projection pumping is a hydrodynamic mechanism, whereby retracting cell-projections inject their cytoplasmic contents into adjacent cells [3]. A recent independent report supports the novel cell-projection pumping mechanism and relates chemotherapy resistance of multiple myeloma cells to the uptake of mitochondria from co-cultured bone marrow-derived stromal cells [21].

An outline of cell-projection pumping is provided because the mechanism has only recently been described [3]. In brief, most cells extend and retract cell-projections as part of their normal function. Increased hydrodynamic pressure in retracting cell-projections normally returns cytoplasm to the cell body. In cell-projection pumping, the cytoplasm, in retracting cell-projections, partially equilibrates into adjacent recipient cells via micro-fusions that form temporary inter-cellular cytoplasmic continuities. We have demonstrated cell-projection pumping by fluorescence confocal time-lapse microscopy, as well as by time-lapse holotomography, and have also published supporting high resolution 3D confocal images [3]. To explore cell-projection pumping, we combined mathematical modelling with comparison of the predictions from the model with experimental results and computer simulations based on experimental data [3]. The mathematical model predicts preferential cell-projection pumping into cells with lower cell stiffness, expected from equilibration of pressure towards least resistance. Predictions from the mathematical model are satisfied when human dermal fibroblasts (HDFs) are co-cultured with SAOS-2 osteosarcoma cancer cells and fluorescence exchange related with cell stiffness by atomic force microscopy. When the transfer into 5000 simulated recipient SAOS-2 cells or HDFs was studied in computer simulations, inputting experimental cell stiffness and donor cell fluorescence values generated simulated transfers to simulated recipient cells similar to those seen by experimenting. The published cell-projection pumping mechanism provides a reasonable basis for further exploration of the biological effects [1,3], as in the current paper.

Our time-lapse microscopy observations show that cell-projection pumping is the main mechanism for the transfer of fluorescently labelled HDF contents into co-cultured SAOS-2 cells. More modest transfers were also seen from SAOS-2 cells into HDFs. Mathematical modelling and computer simulations demonstrated that this mechanism could account for all observable fluorescence transfers in this culture system [3]. Tunnelling nanotubes, exosomes or phagocytic uptake of fibroblast fragments are inconsistent as possible mechanisms to account for our microscopy observations [3]. For this reason, it is convenient to study co-cultures of HDFs with SAOS-2 cells for the purpose of exploring the biological significance of cell-projection pumping. 

A morphological analysis of SAOS-2 cells in co-cultures with human gingival fibroblasts revealed reduced cell circularity and increased cell-profile area in SAOS-2 cells that had received a fluorescent fibroblast label [1]. When SAOS-2 cells were separated from co-cultured fibroblasts by transwell membranes, fibroblast cytokine synthesis was markedly different to when cell-projection pumping was permitted by direct contact [31]. Furthermore, SAOS-2 cells co-cultured with fluorescently labelled HDFs could be separated by fluorescence-activated cell sorting (FACS) into those with high and low levels of HDF label uptake [2]. Co-cultured SAOS-2 cells with high levels of HDF label and separated from their fellows by FACS had increased cell migration in scratch assays [2]. In addition, a FACS analysis demonstrated that SAOS-2 cells with high levels of HDF fluorescent label had increased cell size and increased internal structural complexity and both these changes are consistent with the notion that cell-projection pumping transfers fibroblast organelles and cytoplasm into SAOS-2 cells [2]. 

The emergence of cancer cell diversity is considered a critical factor for cancer progression and resistance to treatment [13,32]. While outgrowth of genetically distinct sub-clones of cancer cells is a key driver for cancer cell diversity, interactions of cancer cells with stromal cells in the surrounding tumour microenvironment are increasingly thought to be important [4,6,7,13,20,21,32,33,34,35]. Considered in this light, our data exploring SAOS-2 phenotypic changes following receipt of fibroblast contents [1,2,3,31] suggest that cell-projection pumping may contribute to clinically important cancer cell diversity and the tumour microenvironment, where SAOS-2 morphological changes [1,2] are relevant to histopathological diagnosis [36], altered cytokine synthesis [31] contributes to the tumour microenvironment [33,35] and increased SAOS-2 migration [2] contributes to cancer spread through the body [32,37]. Although we have focused on fibroblast co-cultures with SAOS-2 cells, we observed similar transfers and phenotypic changes when fibroblasts were co-cultured with other cancer cell lines, including cell lines from melanomas, colon carcinomas, ovarian carcinomas, lung cancer and osteosarcomas [1,2]. 

While our published work is informative on cell-projection pumping and some aspects of the possible significance for cancer, we have been considering limitations of the methods we used. One major difficulty was that the changes in the SAOS-2 phenotype were related to levels of fibroblast fluorescence uptake, but any fibroblast label a given SAOS-2 cell may have received was effectively halved when the cell divided. The effect of this on FACS-separated SAOS-2 populations was to shift the daughter cells of SAOS-2 cells that may have had high levels of fibroblast label into the ‘low fibroblast label’ population, thus undermining the assays. Although the results for cell proliferation in FACS-separated co-cultured SAOS-2 were negative, it occurred to us that increased SAOS-2 cell proliferation could have been masked by such ‘fluorescence halving’ and we felt unable to make a clear conclusion on whether cell-projection pumping affected SAOS-2 cell division [2]. In addition, we saw that our earlier morphological analysis of co-cultured SAOS-2 cells in fixed monolayers [1] would have been similarly affected by SAOS-2 cell division and also provided no information on the discrete history of individual cells. 

Now, we address these limitations by single-cell tracking of SAOS-2 cells co-cultured with HDFs. This method records the behaviour and fate of individual cells and their progeny across multiple cell divisions and generations [38,39,40]. By identifying the fate of individual cells, single-cell tracking overcomes the limitations of pooled cell assays that average the outcomes for thousands of cells. An interesting finding of single-cell tracking studies is that sister cells are more similar to each other than they are to their mother cell or their own progeny [38,39,40].

It was especially interesting for us to explore the possibility that the memory of the HDF cytoplasm received by SAOS-2 mother cells might be preserved in daughter cells. In this study, we assess the effect of cell-projection pumping from HDFs into SAOS-2 cells on cell-profile area, cell circularity, cell migration velocity and proliferation. We related the SAOS-2 phenotype to the absolute levels of fluorescence acquired from HDFs, as well as to what we term ‘compensated fluorescence’, which assigns, to each paired daughter cell, half of their mother cell’s fluorescence which is lost to each sister on mother cell division. Taking the experience of mother cells into account in this way and comparing results for absolute with compensated florescence, we were able to deduce if phenotypic effects were inherited by their daughters, or if mitosis reset SAOS-2 cells to their native state. We further related divergence in the phenotype of paired sister cells to differences in the levels of HDF contents received. 

## 2. Materials and Methods

### 2.1. Materials

All cell culture media, phosphate buffered saline (PBS), trypsin (0.25%)/EDTA (1 mM) and bovine calf serum (BCS) were purchased from Thermo Fisher Scientific (Waltham, MA, USA). Gelatin was from Sigma-Aldrich (St. Louis, MO, USA). Tissue culture plasticware was purchased from Costar (Cambridge, MA, USA). CSL Biosciences (Parkville, VIC, Australia) supplied antibiotics penicillin and streptomycin. ICN Biomedicals Inc. (Costa Mesa, CA, USA) provided amphotericin B. SAOS-2 osteosarcoma cells (HTB-85) and HDFs (HDF-616) were from the American Type Culture Collection (Manassas, VA, USA). The lipophilic fluorescent probes DiD (excitation 644 nm, emission 665 nm) and DiO (excitation 484 nm, emission 501 nm) Vybrant cell labelling solutions were from Molecular Probes, Life Technologies (Grand Island, NY, USA). The black-bottomed 24-well plates were from Ibidi (Gräfelfing, Barvaria, Germany). 

### 2.2. Cell Culture

The antibiotics penicillin (100 U/mL), streptomycin (100 µg/mL) and amphotericin B (2.5 µg/mL) were used throughout all cell culture. HDFs were always cultured on gelatin-coated surfaces (0.1% in PBS) in DMEM (15% BCS). SAOS-2 cells were cultured in DMEM with BCS (10%). Cells were harvested using trypsin-EDTA into BCS to neutralize trypsin and pelleted by centrifugation before passage at a ratio of 1 to 3. All cell cultures were performed at 37 °C under CO_2_ (5%) and at 100% humidity. 

### 2.3. Labelling of Cells with Lipophilic Fluorescent Membrane Markers

Labelling solutions of DiD (1 mM) and DiO (2 mM) were prepared in DMEM with 10% BCS and applied to cells for 1 h. Then, cells were washed twice with PBS before overnight culture with DMEM with BCS (15%), followed by two further washes with PBS in order to ensure removal of any unbound label [1,2,3].

### 2.4. Co-Culture Conditions

All experiments were performed with cells cultured in gelatin-coated (0.1% in PBS) black-bottomed 24-well culture plates. HDFs were seeded from 1 to 2 × 10^4^ cells per cm^2^ and allowed to adhere overnight before labelling with DiD and further overnight culture in DMEM with BCS (2%), as outlined above. SAOS-2 cells were seeded prior to labelling at near confluence in M199 with BCS (15%) and allowed to adhere overnight before labelling with DiO and further overnight culture in DMEM with BCS (2%), as outlined above. Then, pre-labelled SAOS-2 cells were seeded over HDFs in DMEM with BCS (10%) at a culture density of 4 × 10^4^ cells per cm^2^ for a co-culture of up to 5 days, with experiments terminated as monolayers approached confluence. Seven separate experiments were conducted and coded from ‘a’ to ‘g’. One experiment was conducted over 2 days (experiment b), one over 3.6 days (experiment c), four over 4 days (experiments a, d, e and g) and one over 5 days (experiment f). The media were changed on day 3 for experiments extending 4 and 5 days. Control cultures comprised HDFs and SAOS-2 cells labelled and seeded in parallel for isolated cell culture. 

### 2.5. Time-Lapse Recordings

Experimental culture plates were placed in a humidified culture chamber that was mounted on a Leica DM 16000B fluorescence and phase-contrast microscope and maintained under CO_2_ (5%) at 37 °C. Viewing cells through a 20× objective, Micro-Manager open source software [41] was used to construct 3 × 3 grids of contiguous visual fields, each measuring 643 μm × 482 μm. A slight overlap of adjacent visual fields was made to avoid potentially disruptive gaps, so that grids measuring approximately 1920 μm × 1440 μm were observed in each well. Recordings were made of fluorescently labelled co-cultured cells, as well as fluorescently labelled HDF and SAOS-2 control cells cultured alone. The recording of images was controlled by Micro-Manager software, using a Leica DFC365 FX camera. Phase-contrast images were collected at 15 min intervals. Fluorescence images were acquired at a lower rate of one image per 4 h, to minimize photobleaching and phototoxicity. Final recordings were assembled into coherent time-lapse sequences using code developed in MATLAB (MathWorks, Natick, MA, USA). 

### 2.6. Single-Cell Tracking and Analysis

Single-cell tracking software was adapted from a version developed in MATLAB and, which was subsequently made publicly available as TrackPad: (https://github.com/Jamcor/TrackPad, accessed on 14 December 2021). This was used to follow the fate of individual cells and their progeny in phase-contrast time-lapse recordings [38,39]. Cells present at the beginning of experiments were defined as the ‘starting population’, while sequential cell divisions generated ‘first’, ‘second’, ‘third’ and very occasionally ‘fourth’ generations of progeny. The number of cells tracked in each experiment is given for co-cultures of HDFs and SAOS-2 cells (Appendix A) and in control cells cultured in isolation (Appendix A). Across all experiments, 1846 co-cultured SAOS-2 cells were tracked from 607 starting cells, 992 co-cultured HDFs were tracked from 523 starting cells, 1514 control SAOS-2 cells were tracked from 540 starting cells and 849 control HDFs were tracked from 458 starting cells. From this, a total of 5201 cells were tracked in the current study. Details of how precise progeny relationships were coded are given in the ‘Explanatory Notes’ for Appendix A—an Excel spreadsheet with all experimental data summarised. 

The ultimate fate of all tracked cells was determined as either, cell division, incomplete division, apoptosis, or ‘incomplete’ (meaning the cell was either lost from view or reached the end of the experiment). Discrete progeny relationships among all tracked cells were unambiguous. Individual cells were segmented for the analyses of cell circularity and cell-profile area only at times in which both phase-contrast and fluorescence images were available, that is, only at multiples of 4 hourly intervals. Where a tracked cell either divided, underwent apoptosis, failed to divide, or was lost to the field of vision, its phase-contrast image was segmented at the time of the immediately prior fluorescence image. The cell position in the x–y position of the image was recorded, as well as the time of phase-contrast and fluorescence image capture, cell arrival on mother cell mitosis, cell apoptosis, loss of view of the cell, or mitosis. Because the mitotic arrival of cells in the starting generation was undefined, it was not possible to determine intermitotic time nor the time till apoptosis for starting-generation cells. 

### 2.7. Segmentation of Cells and Dependent Calculations for Cell Circularity, Absolute Fluorescence and Compensated Fluorescence

Cells were manually segmented from phase-contrast images, giving results for cell-profile area and cell peripheral circumference. Cell circularity was calculated from these by the equation cell circularity = 4pi(Cell profile area)/(cell peripheral circumference)^2^. DiD fluorescence was quantitated for segmented cells by the summation of intensity of red pixels in fluorescence images, masked by the shape of the segmented cells. 

Data were imported into RStudio open source software [42]. We were interested in comparing the relationship of the SAOS-2 cell phenotype with the levels of DiD fluorescence accepted from co-cultured HDFs. Further, we were interested in considering if phenotypic effects were carried post-mitosis into later generations of cells. To explore this possibility, the SAOS-2 cell phenotype was related to two separate measures for receipt of HDF fluorescence. The absolute fluorescence of cells (Fa) comprised the DiD fluorescence observed in SAOS-2 cells. The compensated fluorescence from the mother cell (Fmc) was calculated for tracked cells that arose by mitosis during the experiment. This awarded, to each of the two sister cells, half of the mother cell’s fluorescence, thus compensating for the distribution of mother cell fluorescence to both daughters. It was assumed that cell division distributed mother cell fluorescence equally to both daughters, so that the numeric correction comprised the addition, to the Fa of each daughter, of half of the immediate mother cell’s Fa value. Separate preliminary analyses compensating for fluorescence from all ancestor generations of cells was also attempted, but it was realized that the variance inherent to the measurements made exceeded the levels of accuracy needed for those calculations to have meaning. 

### 2.8. Determination of Cell Migration Velocity

Moving cell location was determined during single-cell tracking in Trackpad, while further analyses were performed in RStudio, similar to earlier reports [38,39]. In brief, the mean cell migration velocity was calculated for all tracked cells, by firstly identifying cell centroids in phase-contrast images at relevant time points at 2 h intervals, performed in Trackpad. The distances migrated between centroids and time points were calculated in RStudio in a cartesian plot, that is, by the square root of the summated squares of differences for horizontal and vertical coordinates. All distances that a given cell had travelled were then summated and divided by the total time the cell was tracked. Please note that the 2 h interval was selected on the basis that the distances travelled for shorter time intervals were often within the range of error for microscope stage relocation. The last time point frequently did not coincide with the 2 h interval and, where the last time interval was less than 2 h, the preceding time point was removed to create a final time interval greater than 2 h. 

### 2.9. Normalization of Fluorescence Values 

To aid the comparison across experiments, despite the variability inherent to DiD fluorescence labelling, Fa and Fmc values for all SAOS-2 cells were normalized relative to the median DiD fluorescence of co-cultured SAOS-2 cells in the starting generation, which was defined as having a value of 100 normalized fluorescence units. 

### 2.10. Correlation by Kendall’s Tau of Cell-Profile Area, Cell Circularity and Cell Migration Velocity with Receipt of HDF Fluorescence

Cell-profile area, cell circularity and migration velocity of individual SAOS-2 cells were compared separately against both Fa and Fmc. Within individual experiments, the correlation was assessed by Kendall’s tau and its statistical significance was noted. Kendall’s tau was determined for individual generations of cells within experiments, as well as for sequentially added generations, to both summarize results and examine possible confounding effects from prolonged culture or cell crowding. 

### 2.11. Correlation of Mitosis with Receipt of HDF Fluorescence

Cells were identified as either undergoing mitosis or not and, because these were ‘statistically nominal data’, it was not possible to use Kendall’s tau to examine the association between receipt of fluorescence and SAOS-2 cell division. Instead, the receipt of fluorescence by SAOS-2 cells that subsequently underwent division was directly compared with that in cells that did not experience mitosis. It was inherent to the culture system that mitosis was asynchronous and that the times that dividing and non-dividing cells arose varied greatly, as did the times for which these cells were observed. To account for this, values for Fa and Fmc were divided by the time of observation, so that it was the rate of fluorescence uptake (Fa/day and Fmc/day) that was examined when considering the effects of receipt of HDF fluorescence on SAOS-2 cell division. 

### 2.12. Calculation of an Index for Persistence of Phenotypic Effect Inherited from Mother Cells

An ‘index of persistence’ was calculated to assess inheritance, by daughter cells, of any phenotypic effects of fluorescence uptake observed in their respective mitotic mother cells. In this index, a value of ‘0’ indicated no evidence of persistence and ‘1’ indicated strong evidence of persistence of the effect. Please note that the calculated persistence index did not relate to the strength of persistence of phenotype into descendent cells. Instead, the index related to the proportionate number of experiments where evidence of persistence was seen.

The rationale for calculation of the persistence index for cell-profile area, cell circularity and cell migration velocity was that, if there was persistence of effect beyond mitosis, disorder would be introduced to any correlation of Fa with the phenotypic effect, thus reducing the strength of Kendall’s tau. However, the numerical compensation of fluorescence in Fmc would improve order, thus increasing the strength of Kendall’s tau. On the other hand, in the absence of persistence of the effect beyond mitosis, the numerical compensation of fluorescence would introduce disorder for correlation with Fmc, reducing the strength of Kendall’s tau relative to that for Fa. From this, by comparing Fa with Fmc, it was possible to deduce whether or not the effect of receiving HDF fluorescence on the phenotype of mother cells survived mitosis to be inherited by the two daughter cells. For individual experiments, a value of +1 was assigned where tau for Fa < tau for Fmc, 0 was assigned where tau for Fa = tau for Fmc and -1 was assigned where tau for Fa > tau for Fmc. Where negative correlations were involved, all tau values were multiplied by −1 to maintain comparability of the final results. Then the results of these comparisons for all experiments were summated and averaged to yield scores ranging from −1 to 1. The final persistence index values ranging from 0 to 1 were calculated from these averaged scores as persistence index = (1 + Final Score)/2. 

Because the data on mitosis were unique amongst other recordings made, being ‘statistically nominal’, a modified approach was required to calculate the persistence index for the effect of receiving HDF fluorescence on SAOS-2 cell division. The values for Fa and Fmc in dividing cells were first divided by those in non-dividing cells, to yield the ‘proportionate fluorescence values’ pFa and pFmc for each experiment. A pFmc higher than pFa, was interpreted as evidence of persistence with assignation of a score of +1; a pFmc lower than or equivalent to pFa was interpreted as evidence against persistence of effect, with assignation of a score of −1 for the individual experiment. Further summation, averaging and final calculation of persistence index were as described above for other phenotypic features. 

### 2.13. Analysis of the Relationship between Acquired Fluorescence and Phenotype in Paired SAOS-2 Sister Cells 

It is logical to expect that, where there is clear effect on SAOS-2 phenotype of uptake of HDF fluorescence, that there would be correlation when the differences between paired sister cells in phenotype are plotted against differences in fluorescence acquired by the paired sister cells. Sister cells, by definition, can only be identified from generation 1 onwards and Kendall’s tau was determined for differences between paired sister cells in cell-profile area, cell circularity, cell migration velocity and inter-mitotic time. 

### 2.14. Evaluation of Statistical Significance

Statistical significance was accepted for *p* < 0.05. Statistical significance of divergence of Kendall’s tau from an expected correlation of 0 was evaluated by the one-sample Wilcoxon test. Statistical significance of differences in Kendall’s tau between Fa and Fmc for all experiments, as well as that of differences in phenotype between paired groups of cells, was evaluated by the Wilcoxon signed-rank test. The Mann–Whitney U test was used to compare unpaired results. While most statistical evaluations were performed in RStudio, it was occasionally convenient to perform analyses using Prism software (9.2.0; Graphpad Software, San Diego, CA, USA). 

## 3. Results

### 3.1. There Was Marked Transfer of HDF Fluorescent Label to SAOS-2 during Co-Culture 

HDFs and SAOS-2 cells labelled clearly with DiD and DiO fluorescent markers and, when these cells were co-cultured, there was an appreciable and highly localized transfer of DiD from HDFs to SAOS-2 cells, typical of that expected for cell-projection pumping (Figure 1). 

Cell migration and division were readily seen in phase-contrast time-lapse recordings (Appendix A), while morphological changes with regard to cell-profile area and cell circularity were also readily appreciated in both phase-contrast and fluorescence images (Figure 2). The accumulation of HDF fluorescence in many but not all SAOS-2 cells was visually apparent in fluorescence time-lapse recordings (Figure 2; Appendix A).

### 3.2. SAOS-2 Cells Had Lower Cell-Profile Area and Migration Velocity and Higher Cell Circularity than HDF 

Although the cell-profile area of control SAOS-2 cells and HDFs cultured in isolation varied both within and among experiments, SAOS-2 cells were consistently different compared with HDFs (Appendix A). The median value of the cell-profile area in SAOS-2 cells cultured in isolation was 1359 μm^2^, which was significantly lower than that in HDFs in isolated cell culture (median value 4179 μm^2^), while cell circularity in SAOS-2 cells was higher than in HDFs (SAOS-2 cells, median of 0.79; HDFs, median of 0.35, *p* < 0.016; Wilcoxon signed-rank test). Separately, HDFs had significantly faster cell migration than SAOS-2 cells (HDFs, median of 316 μm/day; SAOS-2 cells, median of 84 μm/day; *p* < 0.016, Wilcoxon signed-rank test). These general differences between SAOS-2 cells and HDFs in cell-profile area, circularity and migration velocity were retained when the cells were co-cultured (*p* < 0.016, Wilcoxon signed-rank test).

### 3.3. SAOS-2 Cell-Profile Area Correlated with Receipt of HDF Fluorescence and the Effect Did Not Persist Post-Mitosis

Figure 3 shows typical results, in this case, from experiment ‘a’, with strong correlation between cell-profile area in co-cultured SAOS-2 cells and both Fa and Fmc for all generations of cells tracked. In addition to considering individual generations of cells, it was convenient to summarize the effects by examining the correlations when successive generations of cells were analysed together. As shown in Figure 3, for example, the starting and first generations were considered as a single group to yield one pair of Kendall’s tau values for Fa and Fmc, while a further pair of tau values was obtained when all generations were considered together. 

The results were similar for all experiments (Table 1) and, when the cells of all generations were considered together, the correlation between receipt of HDF fluorescence and SAOS-2 cell-profile area was stronger for Fa than for Fmc (*p* < 0.016, Wilcoxon signed-rank test). This was generally retained when cells across earlier sequential generations were considered (Appendix A). Despite this correlation, there was no clear difference in cell-profile area between SAOS-2 cells in co-cultures and those in isolated control cultures (Appendix A). The reduced correlation in Fmc relative to Fa was visually reflected in scatter plots, by an increasing spread of data points (Figure 3). There was no evidence of persistence of this effect on phenotype in daughter cells, after mother cell division (Table 1).

There was correlation between the differences in Fa of paired sister cells and difference in cell-profile area of these cells (Figure 4; Appendix A; *p* < 0.032, one-sample Wilcoxon test considering generations 1 and 2 together). 

### 3.4. Cell Circularity in Co-Cultured SAOS-2 Cells Was Inversely Correlated with Receipt of HDF Fluorescence and the Effect Persisted Post-Mitosis

Figure 5 shows the results from experiment ‘a’, where there was weak although statistically significant inverse correlation between Fa and cell circularity in co-cultured SAOS-2 cells. The inverse correlation was also seen between cell circularity and Fmc. This correlation was weaker when all generations were considered together (Figure 5), as was generally the case across most experiments (Appendix A); this likely reflected the confounding effects of cell crowding at later time points. For this reason, the analysis across experiments was performed on the grouped results of the starting and first generations (Table 2). 

The strength and statistical significance of the inverse correlation was higher for Fmc than for Fa in four experiments (Table 2; Appendix A). However, the reverse was seen in two experiments (Table 2; Appendix A), while, in one experiment, the correlation seemed reversed. Modest persistence of the effect past mother cell division was apparent from the calculated persistence index of 0.71. SAOS-2 cells had lower cell circularity in co-culture compared with controls in five of seven experiments (experiments b, c, e, f and g), reaching statistical significance in three of these (*p* < 0.025, Mann–Whitney U test; Appendix A). 

Appendix A shows that, in only two experiments (e and g), any correlation was seen in the differences between sister cells for cell circularity and acquired fluorescence. In keeping with the correlations between Fa and Fmc with cell circularity, this correlation too was weak and was only statistically significant when cells from all generations were considered together (experiment e, Kendall’s tau = 0.12, *p* < 0.05; experiment f, Kendall’s tau = 0.14, *p* < 0.035). 

### 3.5. SAOS-2 Cell Migration Velocity Had Weak Correlation with Receipt of HDF Fluorescence and the Effect Did Not Persist Post-SAOS-2 Cell Division

Weak correlation was seen between Fa and cell migration velocity, but this did not reach statistical significance in some experiments (Figure 6, Table 3; Appendix A). Similar correlation was seen for Fmc, but this was less often statistically significant and occasionally reversed (Table 3). SAOS-2 cell migration velocity was greater in co-cultures compared with control cultures in six of seven experiments (experiments b, c, d, e, f and g), but this only reached statistical significance in three experiments (*p* < 0.035, Mann–Whitney U test; Appendix A). There was negligible evidence of persistence beyond cell division of this modest phenotypic effect (Table 3). In addition, there was no correlation for differences between paired sister cells in migration velocity and fluorescence uptake (Appendix A). 

### 3.6. Increased HDF Fluorescence Transfer to SAOS-2 Cells during Co-Culture Was Associated with Subsequent SAOS-2 Cell Mitosis and There Was No Evidence of Persistence of This Post-Cell Division

Mitosis was frequently seen amongst tracked cells (622 of 1846 co-cultured SAOS-2 cells; 235 of 992 co-cultured HDFs; 488 of 1514 control SAOS-2 cells cultured in isolation; 198 of 849 control HDFs cultured in isolation) to produce sequential generations of cells (Appendix A). Apoptosis was much less common (114 of 1846 co-cultured SAOS-2 cells; 18 of 992 co-cultured HDFs; 98 of 1514 control SAOS-2 cells cultured in isolation; 13 of 849 control HDFs cultured in isolation) and did not appear to contribute significantly results (Appendix A). 

To evaluate the effect of receiving HDF fluorescence on SAOS-2 cell mitosis, co-cultured SAOS-2 cells from all generations were considered together, excluding late generations where there was either insufficient experimental time for cell division to occur, or where there was an insufficient number of dividing and/or non-dividing cells for proper comparison. Cells that subsequently underwent division had generally higher rates of both Fa and Fmc uptake than those that did not (Figure 7, Table 4). There was negligible evidence of persistence of the effect post-mitosis (Table 4).

## 4. Discussion

Findings of the current work are broadly consistent with the work of others who report increased cell migration and/or proliferation following transfer of cell contents via tunnelling nanotubes [7,22,23,24,25,26], as well as with our own earlier reports [1,2]. Although special interest has been shown in the literature for the role of mitochondria [4,6,7,20,21,22,23,24,25,26,27], our observation of bulk cytoplasmic transfer via cell-projection pumping, including of cytoplasmic protein, plasma membrane alkaline phosphatase and organelles smaller than mitochondria [1,2,3], makes us cautious in focusing on this single organelle as the critical factor for all changes observed. Instead, we suggest that, in addition to mitochondria, any cellular component transferred has the potential to profoundly affect the recipient cell. Nonetheless, we can report that we have seen, in separate work still unready for publication, that HDF mitochondria seem transferred to SAOS-2 cells in an appreciable amount by cell-projection pumping. 

One reason for using single-cell tracking in the current study is the interest in exploring whether or not the phenotypic impact of fibroblast transfers to cancer cells, transcended mitosis to affect daughter cells. A particular challenge to addressing this question was that post-mitotic SAOS-2 cells were open to receive further transfers from HDFs, obscuring any phenotypic effects inherited from mother cells. It seemed reasonable for us to assume that receipt of HDF fluorescence was a good proxy for the amount of HDF contents received and phenotypic effect in mother cells and that there was equal distribution of mother cell fluorescence to each mitotic daughter cell. On that basis, our approach of comparing correlation of phenotype against Fa with that for Fmc, to deduce persistence or otherwise of phenotypic effects post-mitosis, also seemed reasonable.

At the level of individual experiments, it was possible to make a direct comparison of Kendall’s tau for Fa and Fmc for cell-profile area, cell circularity and cell migration velocity and of pFa with pFmc with regard to cell division. However, there was some variability in outcomes of experiments; therefore, in addition to the evaluation with statistical tests appropriate to include multiple experiments, it was helpful to determine an index of persistence. We found no clear precedent for our approach in the literature and we suggest it may find application elsewhere by others. 

It seemed very clear that associations between receipt of HDF fluorescence and SAOS-2 cell-profile area, migration velocity and mitosis, were reset by cell division. This suggests that whatever HDF cell-projection pumping delivers to SAOS-2 cells, which drives these phenotypic changes, is either degraded or otherwise unstable, so that the effects do not survive mitotic events. This is consistent with separate reports on the importance of scheduled protein and RNA degradation, associated with cell cycle progression, suggested by others as important for resetting cells after mitosis [43,44,45,46,47,48,49,50,51]. In addition, because mitochondrial function is retained post-mitosis [52], the current findings reinforce our doubt about the possible importance of mitochondria in accounting for the changes we describe.

However, with regard to cell circularity, a degree of persistence was observed. It might be thought that the modest persistence observed was artifactual, due to the low levels of correlation involved. However, this would be inconsistent with the absence of persistence for cell migration velocity, where correlation was even weaker. The persistence of circularity effects post-mitosis could reflect epigenetic changes in SAOS-2 cells, but, given the variability across experiments, it is more likely that there is degradation of the driver or drivers responsible for cell circularity similar to those for other phenotypic effects studied, but that these are sufficiently robust to penetrate post-mitosis into daughter cells. Clarity awaits characterization of the precise contents transferred from HDFs and of the mechanisms responsible for effecting these phenotypes. 

Separately, we exploited the high similarity of sister cells [38,39,40], to help verify phenotypic effects of fluorescence transfers seen. It was reassuring to see the expected correlation in this for cell-profile area. Although divergence in cell-circularity for sister cells was only seen in two experiments, this was perhaps to be expected, given the much weaker level of correlation involved. Similar arguments apply to the absence of correlation between paired sister SAOS-2 cells with regard to cell migration velocity, where the direct correlation with Fa and Fmc was even weaker than for cell circularity. 

A further motivation for using single-cell tracking was to overcome potential confounding effects of pooled cell assays observed in earlier works [1,2]. In this regard, the outcomes of the current study are mixed. Of benefit was that the current study, for the first time, associated receipt of fibroblast label with SAOS-2 cell mitosis. We earlier used FACS to separate SAOS-2 cells with high levels of HDF fluorescence uptake from those with low uptake. In that work, we found no proliferative effect [2], which, at first light, seems inconsistent with the current findings by single-cell tracking. However, since we demonstrate, in addition to the pro-mitotic association, the failure of this to persist in daughter cells, we see that these two studies can be reconciled considering the distribution through the FACS column of daughter cells with high levels of inherited mother cell fluorescence and the pooled cell proliferation assay that extended over several days and cell divisions [2]. It appears that mitosis can be driven, while SAOS-2 cells have the opportunity to receive new transfers from fibroblasts during actual co-culture, but that, once isolated from the fibroblasts, as for example by FACS separation, the cell-projection pumping driver for mitosis is lost. 

The single-cell approach had the further benefit, over our initial study of fixed monolayers of cells, where increased cell-profile area and reduced cell circularity were seen, but only at the cell population level as shifts in the data for SAOS-2 cells that had clear fluorescence uptake, relative to those that did not [1]. The current demonstration of changes in cell circularity and cell-profile area at the single-cell level increases confidence in our earlier findings [1]. This is especially the case for the cell-profile area, where data from paired sister cells were particularly convincing. We previously demonstrated by FACS analysis that the uptake of HDF contents by SAOS-2 cells correlates with increased size of SAOS-2 cells [2] and it is tempting to imagine that the current correlation of fibroblast fluorescence transfer with increased SAOS-2 cell-profile area is a simple expression of that. However, cell-profile area in two-dimensional culture is more reflective of active cell stretching than of mere volume, so we suspect this observation indicates a change in SAOS-2 cell activity rather than passive cell size. 

In as much as the current single-cell tracking study had advantages over previous pooled cell assays [1,2], there were also some disadvantages. We found that data were sensitive to culture density effects, with, for example, data on cell circularity and cell migration velocity from late generations of cells being affected by increased cell crowding. 

We were particularly surprised to find only a slight correlation between receipt of HDF fluorescence and SAOS-2 cell migration velocity, because, in our earlier work with FACS isolated cells, the effect was marked [2]. We suggest that, in addition to stimulating SAOS-2 cell migration via cell-projection pumping, HDFs are also able to inhibit SAOS-2 cell migration in co-culture. This possibility is supported by the work of others, who report that fibroblasts can inhibit cancer cell migration by both soluble- and contact-dependent mechanisms [53]. 

Further surprising us, was absence of evidence of persistence of increased migration post-mitosis, while our earlier work with FACS isolated cells in scratch assays showed increased migration over many days and clearly over multiple cell divisions [2]. It is possible that there might be a persistence of effect, but that the low levels of correlation observed in the current study make that undetectable by the methods used. Another possibility is the leading edge of migrating cells in scratch assays, comprised starting-generation cells that did not divide but effectively filtered themselves out of the pooled population by migrating into scratches. 

Despite the associations found between cell-profile area, cell circularity and cell migration velocity, there was only a modest difference between control SAOS-2 cells cultured in isolation and those in co-culture. Given the variability inherent to the data and the comparatively weak strength of correlations observed, this is perhaps unsurprising. 

Histopathological cancer-grading systems often include the assessment of the variability in morphology amongst cancer cells, formally described as pleomorphism, with high pleomorphism usually indicating a higher grade and worse clinical outcomes [32]. Notably, cellular pleomorphism is recognized by variability in both ‘cell size’ and ‘cell shape’ as seen in histological sections [32,54], both of which have been assessed in the current study by assessment of cell-profile area and cell circularity. We stress that cell-profile area and cell circularity as currently studied are both the result of active functional processes in cells and reflect, at a minimum, operations of cell adhesion, cell stretching and cell polarity in the horizontal plane. The increase in cell-profile area contrasted with the decrease in cell circularity in the current study and these opposing responses were consistent with separate unpublished work in our laboratory, showing that cell-profile area and cell circularity, as measured in the current work, are independent variables. The current findings on cell-profile area and cell circularity are consistent with our earlier suggestion that cell-projection pumping increases morphological cancer cell diversity and that this has a bearing on cancer diagnosis [1]. 

While the significance of the effect of HDF cell-projection pumping on SAOS-2 cell migration now seems complicated by what may be opposing effects, as outlined above, any increase in cancer cell migration has potential to increase the clinical spread of cancer cells in both local cancer invasion of adjacent tissues and spread to distant sites via lymphatics or the circulation [32]. It is the progressive involvement of ever more previously normal tissues and organs by cancer that is perhaps the most devastating aspect of the disease; therefore, even modestly increased cancer cell migration via cell-projection pumping may have biological significance. 

The growth of cancerous tissue is a key hallmark of the disease [32] and clearly requires proliferation of cancer cells. The association of cell-projection pumping with increased cancer cell mitosis seen in the current study seems inherently important. 

From the above, the current data suggest that the uptake of contents from fibroblasts by cell projection pumping may drive the following clinically important aspects: cancer cell morphological diversity, of relevance to cancer diagnosis, prognosis and treatment planning; increased cancer spread through the body by increased cancer cell migration; and increased growth of cancer by increased cancer cell proliferation. 

Nonetheless, we recognize that the current work only establishes a correlation between transfer of contents from fibroblasts with SAOS-2 phenotypic change and does not provide information on the precise agents transferred between cells, or the precise mechanisms through which phenotypic changes in SAOS-2 cells are mediated by the contents transferred. However, given that cell-projection pumping transfers bulk cytoplasm, including cytoplasmic proteins and organelles, and that there are, in addition, transfers of cell surface markers, including alkaline phosphatase [1,3], it should not be surprising that cell-projection pumping is associated with profound phenotypic effects as earlier reported [1,2,31] and as further documented in the current report.

Our interest in the persistence or otherwise of the effects on SAOS-2 cells of receiving HDF contents and demonstration that most effects are reset by mitosis could distort the perception of the clinical significance of cell-projection pumping. Thus, it seems important to note that, in vivo, fibroblasts are nearly always available to provide fresh contents to newly arrived cancer cells; therefore, irrespective of persistence, cancer cells are always open to the impact of cell-projection pumping, which, on the basis of our data, we believe to be important. Since the cell-projection pumping mechanism has only been recently described [3], we envision the future development of novel anti-cancer therapies that target cell-projection pumping. 

## 5. Conclusions

We conclude that the transfer of HDF contents to SAOS-2 cells via cell-projection pumping is associated with the following: increased SAOS-2 mitosis; modestly increased SAOS-2 migration; increased SAOS-2 cell profile area; reduced SAOS-2 circularity. In addition, we conclude that all these phenotypic changes, other than cell circularity, are reset by SAOS-2 mitosis. We suggest that this indicates an important role for cell-projection pumping in clinical cancer histopathologic diagnosis and progression of the disease. We envision the development of novel anti-cancer therapies targeting cell-projection pumping. 

## Figures and Tables

**Figure 1 biomolecules-11-01875-f001:**
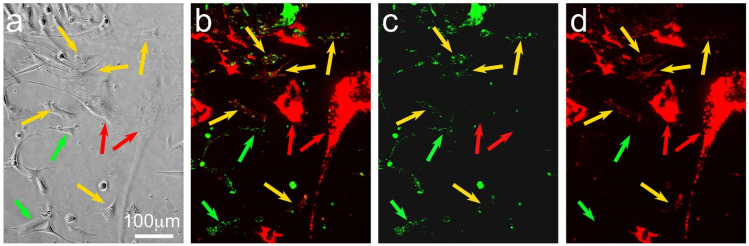
Photomicrographs of a visual field of pre-labelled SAOS-2 cells (fluorescent green marker) co-cultured with pre-labelled HDFs (fluorescent red marker) at the 20 h time point, showing phase-contrast (**a**) and fluorescent images with red and green channels combined (**b**), as well as green (**c**) and red (**d**) channels alone. The lipophilic fluorescent markers were concentrated in organelles, creating a typically punctate appearance (**b**–**d**), so that phase-contrast images (**a**) were more helpful for identification of cell margins. HDFs were clearly red-labelled (red arrows), while SAOS-2 cells were green-labelled. Comparison of combined and separated channels for fluorescence images demonstrates that some SAOS-2 cells had no red HDF fluorescent label (green arrows), while others had received appreciable levels of the red HDF marker (yellow arrows).

**Figure 2 biomolecules-11-01875-f002:**
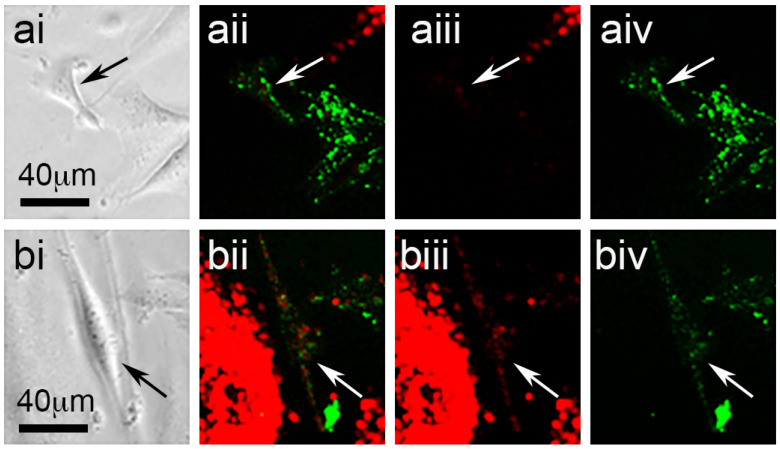
Phase-contrast (**ai**,**bi**) and fluorescence images of a SAOS-2 cell (arrows) in co-culture with HDFs, showing red and green channels together (**ii**), or alone ((**iii**) red and (**iv**) green, respectively), both at the starting time point (**a****i**–**aiv**) and 1.24 days after the first image was recorded (**bi**–**biv**). (**ai**) At the starting time point, the SAOS-2 cell had a triangular form best appreciated by phase-contrast microscopy and had clear green fluorescence without appreciable red fluorescence (**aii**–**aiv**). Measured cell-profile area at the starting time point was 815 μm^2^ and cell circularity was 0.53. (**bi**) At a later time point, the marked SAOS-2 cell presented with a larger cell-profile area of 1483 μm^2^, as well as a more elongated form and a lower circularity value of 0.18. (**bii**–**biv**) The marked SAOS-2 cell also had appreciable red fluorescence acquired from HDFs by the latter time point.

**Figure 3 biomolecules-11-01875-f003:**
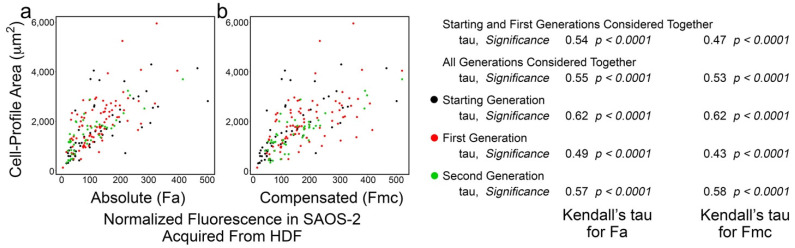
Scatter plots of data from experiment ‘a’ co-culturing SAOS-2 cells with HDFs, showing cell-profile area of co-cultured SAOS-2 cells plotted against normalized fluorescence acquired from HDFs expressed as absolute fluorescence measured (Fa in panel **a**), as well as numerical compensation for halving of fluorescence by cell division in mother cells (Fmc in panel **b**). The generation to which each cell belonged is indicated by the colour. Values for Kendall’s tau of correlation and their statistical significance are shown for individual generations of cells, starting and first generations together and all generations of cells together. (**a**) There was correlation between cell-profile area and Fa. (**b**) This was reduced for Fmc.

**Figure 4 biomolecules-11-01875-f004:**
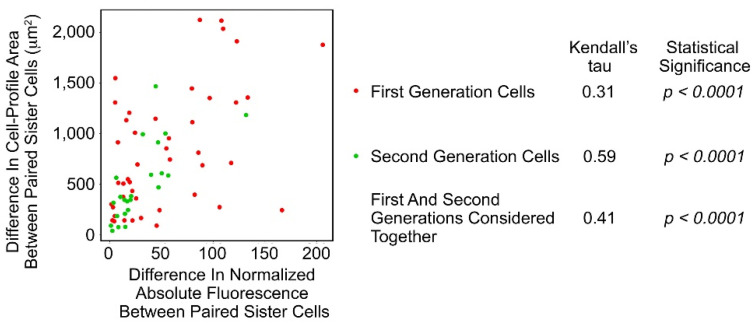
Scatterplot of the differences between paired sister SAOS-2 cells in cell-profile area against differences in fluorescence acquired from co-cultured HDFs. Data are from the same experiment, ‘a’, which is shown in Figure 3. There was correlation in both generations of cells studied, as well as when first and second generations were considered together.

**Figure 5 biomolecules-11-01875-f005:**
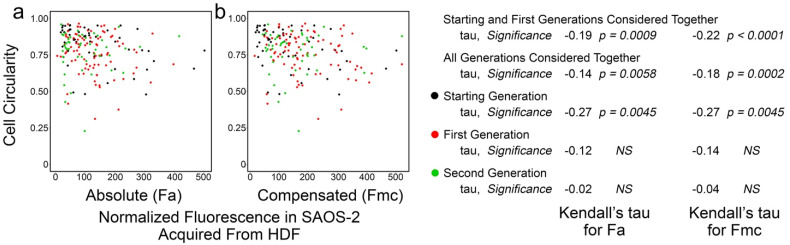
Scatter plots of experiment ‘a’ co-culturing SAOS-2 cells with HDFs, showing cell circularity of co-cultured SAOS-2 cells plotted against normalized fluorescence acquired from HDFs expressed as absolute fluorescence measured (Fa in panel **a**), as well as with numerical compensation for halving of fluorescence by cell division from mother cells (Fmc in panel **b**). The generation to which each cell belonged is indicated by the colour. Values for Kendall’s tau of correlation and their statistical significance are shown for individual generations of cells, starting and first generations together and all generations of cells together. (**a**) There was inverse correlation between cell-profile and Fa. (**b**) This was increased for Fmc.

**Figure 6 biomolecules-11-01875-f006:**
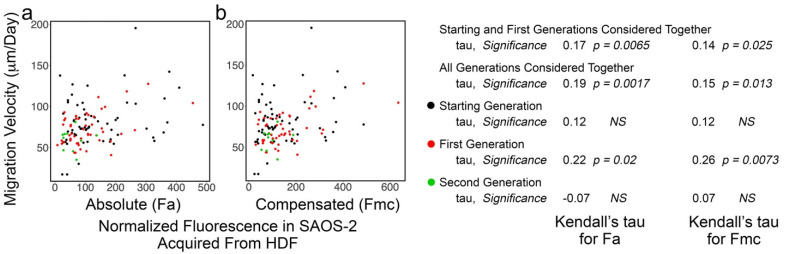
Scatter plots of experiment ‘f’ co-culturing SAOS-2 cells with HDFs, showing cell migration velocity of co-cultured SAOS-2 cells plotted against normalized fluorescence acquired from HDFs expressed as absolute fluorescence measured (Fa in panel **a**), as well as numerical compensation for halving of fluorescence by cell division in mother cells (Fmc in panel **b**). The generation to which each cell belonged is indicated by the colour. Values for Kendall’s tau of correlation and their statistical significance are shown for individual generations of cells, starting and first generations together and all generations of cells together. (**a**) There was weak correlation between cell migration and Fa. (**b**) This was decreased for Fmc.

**Figure 7 biomolecules-11-01875-f007:**
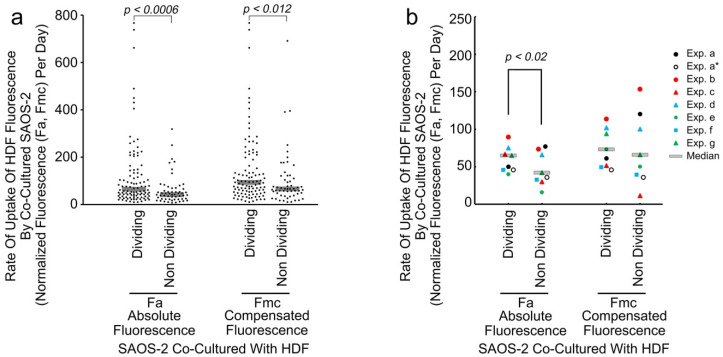
Scattergram showing results of a typical experiment (experiment g) showing the rate of uptake of normalized HDF fluorescence by individual co-cultured SAOS-2 cells, according to whether cells underwent subsequent cell division or not (**a**), as well as median values from all experiments (**b**). Fluorescence is expressed as the rate of uptake of absolute fluorescence measured (Fa/day), as well as with numerical compensation for halving of fluorescence by cell division in mother cells (Fmc/day). (**a**) Dividing cells had clearly higher values for Fa/day than non-dividing cells and this was statistically significant as marked (Mann–Whitney U test). A similar statistically significant difference was seen for Fmc/day. (**b**) Results for all experiments are shown, while experiment ‘a’, which showed atypical results for first-generation cells, is indicated twice—once including first-generation cells and once without (Exp. a*). The median rate of Fa uptake was higher in those cells that subsequently underwent division than in those that did not and this was statistically significant as marked when experiment ‘a’ first-generation cells only were considered, as well as when experiment ‘a’ was excluded from the analyses (*p* < 0.032, Wilcoxon signed-rank test). The relationship for Fmc was similar although less consistent and was not statistically significant when all experiments were considered together.

**Table 1 biomolecules-11-01875-t001:** Kendall’s tau of correlation between cell-profile area of tracked SAOS-2 cells and absolute fluorescence acquired from co-cultured HDFs (Fa), as well as compensation for halving of fluorescence from mother cells by cell division (Fmc). Results for all experiments are shown, considering all generations of cells counted together. Results for other groupings of successive generations of cells are in Appendix A.

	Kendall’s Tau of Correlation between Cell-Profile Area and Receipt of HDF Fluorescence by SAOS-2 Cells	
	Fa	Fmc
*** Experiment a**		
Kendall’s tau of all Generations Considered Together	0.55	0.53
*Statistical Significance of the Above*	*<0.0001*	*<0.0001*
**Experiment b**		
Kendall’s tau of all Generations Considered Together	0.15	0.11
*Statistical Significance of the Above*	*<0.0001*	*<0.0001*
**Experiment c**		
Kendall’s tau of all Generations Considered Together	0.76	0.68
*Statistical Significance of the Above*	*<0.0001*	*<0.0001*
**Experiment d**		
Kendall’s tau of all Generations Considered Together	0.51	0.45
*Statistical Significance of the Above*	*<0.0001*	*<0.0001*
**Experiment e**		
Kendall’s tau of all Generations Considered Together	0.13	0.08
*Statistical Significance of the Above*	*0.001*	*0.04*
**Experiment f**		
Kendall’s tau of all Generations Considered Together	0.55	0.51
*Statistical Significance of the Above*	*<0.0001*	*<0.0001*
**Experiment g**		
Kendall’s tau of all Generations Considered Together	0.54	0.41
*Statistical Significance of the Above*	*<0.0001*	*<0.0001*

Divergence from an expected correlation of 0 was statistically significant for Fa and Fmc (*p* < 0.016) by one-sample Wilcoxon test. There was strong correlation of cell-profile area with receipt of HDF fluorescence. Correlation with Fa (median of 0.54) was stronger than for Fmc (median 0.45) in all experiments (*p* < 0.016 Wilcoxon signed-rank test). The calculated index of persistence was 0 and there was no evidence of persistence of the effect of receiving HDF fluorescence past SAOS-2 cell division. * Indicates the experiment shown in Figure 3. Statistical statements are in italics to aid visual interpretation, and exact values for p are given when these were available.

**Table 2 biomolecules-11-01875-t002:** Kendall’s tau of correlation between cell circularity of tracked SAOS-2 cells and absolute fluorescence acquired from co-cultured HDFs (Fa), as well as compensation for halving of fluorescence from mother cells by cell division (Fmc). Results for all experiments are shown, considering starting and first generations together. Results for other groupings of successive generations of cells are in the Appendix A. Statistical significance is given, where *NS* indicates ‘not significant’ at *p* < 0.05. Where statistical significance was approached but not reached, the calculated *p*-value is given (*NS (p-value)*).

	Kendall’s Tau of Correlation between Cell Circularity and Receipt of HDF Fluorescenceby SAOS-2 Cells	
	Fa	Fmc
*** Experiment a**		
Kendall’s tau of Starting and First Generations Considered Together	−0.19	−0.22
*Statistical Significance of the Above*	*0.0009*	*<0.0001*
**Experiment b**		
Kendall’s tau of Starting and First Generations Considered Together	−0.02	−0.05
*Statistical Significance of the Above*	*NS*	*NS (0.063)*
**Experiment c**		
Kendall’s tau of Starting and First Generations Considered Together	−0.35	−0.36
*Statistical Significance of the Above*	*<0.0001*	*<0.0001*
**Experiment d**		
Kendall’s tau of Starting and First Generations Considered Together	−0.24	−0.23
*Statistical Significance of the Above*	*0.0024*	*0.0032*
**Experiment e**		
Kendall’s tau of Starting and First Generations Considered Together	0.10	0.16
*Statistical Significance of the Above*	*NS (0.06)*	*0.0026*
**Experiment f**		
Kendall’s tau of Starting and First Generations Considered Together	−0.12	−0.16
*Statistical Significance of the Above*	*0.061*	*0.012*
**Experiment g**		
Kendall’s tau of Starting and First Generations Considered Together	−0.22	−0.18
*Statistical Significance of the Above*	*<0.0001*	*0.0003*

Divergence from an expected correlation of 0 was statistically significant for Fa (*p* < 0.05) and approached but did not reach statistical significance for Fmc (*p* = 0.063) by one-sample Wilcoxon test. There was weak inverse correlation of cell circularity with HDF fluorescence in all but experiment ‘e’, where the reverse effect was seen. Strength of the inverse correlation across Fa and Fmc varied amongst experiments. Tau values for Fmc were higher than Fa in 4 experiments (a, b, c and f), while the reverse was the case in experiments ‘d’ and ‘g’. Although statistically compelling within individual experiments, with median Fa = −0.19 and median Fmc = −0.18, differences between experiments were such that no statistically significant result could be attributed to the general pattern. Despite variability, data overall suggest moderately strong persistence of circularity from mother cells (index of persistence of 0.71). * Indicates the experiment shown in Figure 5. Statistical statements are in italics to aid visual interpretation, and exact values for p are given when these were available.

**Table 3 biomolecules-11-01875-t003:** Kendall’s tau of correlation between cell migration velocity of tracked SAOS-2 cells and absolute fluorescence acquired from co-cultured HDFs (Fa), as well as compensation for halving of fluorescence from mother cells by cell division (Fmc). Results from all experiments are shown. Results shown within individual experiments are from either all generations of cells together, starting and first generations together, or, where a second generation of cells was present, starting, first and second generations together. Selection of these within experiments was on the basis of the strongest statistical significance, while results for all grouped cell generations are shown in the Appendix A. Note that there were no statistically significant differences in experiments b and d and, here, selection was performed from starting and first generations considered together. Statistical significance is given, where *NS* indicates ‘not significant’ at *p* < 0.05. Where statistical significance was approached but not reached, the calculated *p*-value is given (*NS (p-value)*).

	Kendall’s Tau of Correlation between Cell Migration Velocity and Receipt of HDF Fluorescence by SAOS-2 Cells	
	Fa	Fmc
**Experiment a**		
Kendall’s tau of Starting and First Generations Considered Together	0.10	0.18
*Statistical Significance of the Above*	*NS (0.088)*	*0.0019*
**Experiment b**		
Kendall’s tau of Starting and First Generations Considered Together	0.02	−0.02
*Statistical Significance of the Above*	*NS*	*NS*
**Experiment c**		
Kendall’s tau of Starting and First Generations Considered Together	0.25	0.19
*Statistical Significance of the Above*	*<0.0001*	*<0.0002*
**Experiment d**		
Kendall’s tau of Starting and First Generations Considered Together	0.06	0.04
*Statistical Significance of the Above*	*NS*	*NS*
**Experiment e**		
Kendall’s tau of all Generations Considered Together	0.08	0.06
*Statistical Significance of the Above*	*0.045*	*0.11*
*** Experiment f**		
Kendall’s tau of all Generations Considered Together	0.19	0.15
*Statistical Significance of the Above*	*0.0017*	*0.013*
**Experiment g**		
Kendall’s tau of Starting, First and Second Generations Considered Together	0.11	0.02
*Statistical Significance of the Above*	*0.0069*	*NS*

Divergence from an expected correlation of 0 was significant for Fa (*p* < 0.016) and Fmc (*p* < 0.05) by one-sample Wilcoxon test. Very weak correlation of cell migration with HDF fluorescence was seen in most experiments and this reached statistical significance in 5 of 7 of these (a, c, e, f and g). The correlation was strongest for Fa in all experiments other than in ‘a’, where the strongest correlation was in Fmc. In 1 experiment (b), there was apparent weak inverse correlation for Fmc, but this was not statistically significant. There was no statistically significant difference between Fa (median, 0.10) and Fmc (0.06). Comparison of Fa with Fmc provided negligible evidence of any persistence of the effect beyond cell division, with a calculated persistence index of 0.14. * Indicates the experiment shown in Figure 6. Statistical statements are in italics to aid visual interpretation, and exact values for p are given when these were available.

**Table 4 biomolecules-11-01875-t004:** The fate of SAOS-2 cells with regard to cell division, related to the rate of previous fluorescence uptake from co-cultured HDFs. Median rates of acquisition of fluorescence are given for absolute fluorescence acquired per day (Fa/day) and fluorescence compensating for halving on division of immediate mother cells (Fmc/day). Statistical significance (Stat. Sig.) as per the Mann–Whitney U test is given, where *NS* indicates ‘not significant’ at *p* < 0.05. Where statistical significance was approached but not reached, the calculated *p*-value is shown (*NS (p-value)*). All generations of cells for which adequate data were available were included in the analyses. However, in experiment ‘a’, first-generation cells had atypical results compared with all other experiments, with disproportionate effect on pooled results. For this reason, two results for experiment ‘a’ are shown, one including these cells and another where only cells in the starting generation of experiment ‘a’ were analysed (Experiment a*). To assess persistence of effect, both pFa and pFmc are shown for all experiments other than for experiment ‘a*’, where pFmc has no meaning.

	Median Fa/Day			Median Fmc/Day		
	Dividing Cells	Non-Dividing Cells	pFa (Dividing Cells/Non-Dividing Cells)	Dividing Cells	Non-Dividing Cells	pFmc (Dividing Cells/Non-Dividing Cells)
Experiment a	49.0	76.9	0.64	61.1	120.1	0.51
*Stat. Sig. of Above*	*0.0022*			*<0.0001*		
Experiment a*	45.6	35.6	1.28	45.6	35.6	-
*Stat. Sig. of Above*	*NS*			*NS*		
Experiment b	89.4	73.3	1.22	113.5	153.3	0.74
*Stat. Sig. of Above*	*NS (0.057)*			*NS*		
Experiment c	66.7	29.9	2.23	51.4	11.4	4.51
*Stat. Sig. of Above*	*<0.0001*			*<0.0001*		
Experiment d	75.0	65.8	1.88	102.0	100.1	1.02
*Stat. Sig. of Above*	*NS (0.0535)*			*NS*		
Experiment e	39.9	15.8	2.53	73.0	50.0	1.46
*Stat. Sig. of Above*	*<0.0001*			*0.0041*		
Experiment f	45.6	32.6	1.40	49.1	39.2	1.25
*Stat. Sig. of Above*	*0.0222*			*NS*		
Experiment g	64.9	42.1	1.54	94.0	65.9	1.43
*Stat. Sig. of Above*	*0.0005*			*0.0115*		

Co-cultured SAOS-2 cells that underwent cell division had higher rates of uptake for Fa than SAOS-2 cells that did not divide. (Excluding experiment ‘a’: median Fa/day for dividing cells = 65.8; median Fa/day for non-dividing cells = 37.3; *p* < 0.032, Wilcoxon signed-rank test. Including experiment a*: median Fa/day for dividing cells = 64.9; median Fa/day for non-dividing cells = 35.6; *p* < 0.016, Wilcoxon signed-rank test.) This was also seen in 5 of 6 relevant experiments for Fmc (c, d, e, f and g), although there was negligible difference in experiment ‘d’ and the reverse was seen in experiment ‘b’ (median Fmc/day for dividing cells = 73.0; median Fmc/day for non-dividing cells = 50.0; not statistically significant). Excluding experiment ‘a’, comparison of pFa with pFmc yielded a negligibly low persistence index of 0.17, so that persistence of the association between uptake of fluorescence and cell division in descendent cells was not supported by these data. Statistical statements are in italics to aid visual interpretation, and exact values for p are given when these were available.

## Data Availability

Data for re-analysis of results are provided in the Appendix A.

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
