# Peer review of "Cell-Projection Pumping of Fibroblast Contents into Osteosarcoma SAOS-2 Cells Correlates with Increased SAOS-2 Proliferation and Migration, as well as Altered Morphology"

_biomolecules, 2021, doi:10.3390/biom11121875_

Round 1
Reviewer 1 Report
The manuscript described the results of experiments using the co-culture of the human-derived fibroblasts (HDF) and the human osteogenic SAOS-2 cells. Cells were incubated either with a red fluorescent dye (HDF) or a green fluorescent dye (SAOS-2 cells). Red fluorescence was detected in some SAOS-2 cells co-cultured with HDF, which indicated cell-projection pumping. The authors performed several analyses to study cell-profile area and migration velocity of co-cultured SAOS-2 cells. Kendall’s tau and statistical significance were performed to assess correlation. The authors found that transfer of HDF contents to SAOS-2 via cell-projection pumping drives: increased SAOS-2 mitosis, modestly increased SAOS-2 migration, increased SAOS-2 cell profile area, and reduced SAOS-2 circularity. However, all these phenotypic changes were reset by SAOS-2 mitosis. These findings are interesting and may impact cancer cell phenotype, relevant to cancer histopathological diagnosis.
The manuscript contains a comprehensive introduction to the problem, well-described methodology. Supplementation of the manuscript by images, additionally to statistical counts, that show phenotypic changes during co-culture of HDF with SAOS-2 can enhance the manuscript.
I recommend inserting of images of phenotypic changes of SAOS-2 cells (such as cell circularity, enter to mitosis, and others mentioned in the manuscript) following cell-projection pumping of fibroblast contents additionally to graphs. This will make visible what the authors are counting.Author Response
Please see the attachment

Reviewer 2 Report
In this manuscript, Mahadevan and co-workers presents correlative results between cancer cell (SAOS2) behavior and the amount of fibroblast organellar (and potentially cytosolic) they contain. This manuscript is in line with earlier studies from the same group on cell-projection pumping. While the process is novel and potentially interesting, it’s hard for this review to appreciate the functional relevance of cell projection pumping for this SAOS cells. While it’s good to see correlation between the fibroblast contents and a few cellular processes, authors should add some functional or physiological context to their findings.
Specific concerns:
- As a cell biologist, this reviewer is not entirely convinced about how the organellar transport actually happen in cell projection pumping. Can authors distinguish between whether the SAOS2 cells bite off a bit of the fibroblast projection or the fibroblast actively pump via some conduit between two cells? The payload would be membrane enclosed in the former case and cytosolic in latter. Also, is the organellar payload primarily mitochondria or other organelles are transferred.
- How do authors reconcile a strong positive correlation observed for cell profile area with a mild negative correlation observed for circularity? Do the SAOS2 cells become more elongated and well spread after the transfer? Please provide more example images and tracking montage as figures and videos as supplemental material.
- Authors observe negligible to poor correlation between migration velocity of the cancer cells for most of the experiment. What is the difference between the migration velocity of SAOS2 cells which don’t receive any transfer or grown alone? Are the differences significant.
4. Authors discuss and conclude important roles for cell projection pumping for cancer pathophysiology or developing anti-cancer therapies. But this statement is an exaggeration as authors haven’t even explored causal links between the phenomenon (cellular material transfer) and the observations they make.
In summary, the results presented here are too preliminary and unsubstantiated for the claims authors are making.
Round 2
Reviewer 2 Report
I'm glad that authors have taken the critique from this reviewer in a positive spirit and have addressed all the comments raised earlier on a well detailed response letter and have made several important changes to their initial version. They have also provided supplemental data and movies to support their manuscript. Authors need to add scale bar and time-stamp to the supplemental movie they have added.